# Dynamics of Neutralizing Antibody and T-Cell Responses to SARS-CoV-2 and Variants of Concern after Primary Immunization with CoronaVac and Booster with BNT162b2 or ChAdOx1 in Health Care Workers

**DOI:** 10.3390/vaccines10050639

**Published:** 2022-04-19

**Authors:** Watsamon Jantarabenjakul, Pimpayao Sodsai, Napaporn Chantasrisawad, Anusara Jitsatja, Sasiprapa Ninwattana, Nattakarn Thippamom, Vichaya Ruenjaiman, Chee Wah Tan, Rakchanok Pradit, Jiratchaya Sophonphan, Supaporn Wacharapluesadee, Lin-Fa Wang, Thanyawee Puthanakit, Nattiya Hirankarn, Opass Putcharoen

**Affiliations:** 1Thai Red Cross Emerging Infectious Diseases Clinical Center, King Chulalongkorn Memorial Hospital, Bangkok 10330, Thailand; watsamon.j@chula.ac.th (W.J.); napaporn.cha@chula.ac.th (N.C.); anusara.j@chulahospital.org (A.J.); 6478011830@student.chula.ac.th (S.N.); nattakarn.th@chula.ac.th (N.T.); rakchanok.p@chulahosptal.org (R.P.); supaporn.w@chulahospital.org (S.W.); 2Department of Pediatrics, Faculty of Medicine, Chulalongkorn University, Bangkok 10330, Thailand; thanyawee.p@chula.ac.th; 3Center of Excellence in Pediatric Infectious Diseases and Vaccine, Faculty of Medicine, Chulalongkorn University, Bangkok 10330, Thailand; 4Center of Excellence in Immunology and Immune-Mediated Diseases, Department of Microbiology, Faculty of Medicine, Chulalongkorn University, Bangkok 10330, Thailand; pimpayao.s@chula.ac.th (P.S.); vichaya.r@chula.ac.th (V.R.); nattiya.h@chula.ac.th (N.H.); 5Programme in Emerging Infectious Disease, Duke-NUS Medical School, Singapore 169857, Singapore; cheewah.tan@duke-nus.edu.sg (C.W.T.); linfa.wang@duke-nus.edu.sg (L.-F.W.); 6The HIV Netherlands Australia Thailand Research Collaboration (HIV-NAT), Thai Red Cross AIDS Research Centre, Bangkok 10330, Thailand; jiratchaya.w@hivnat.org; 7Department of Medicine, Faculty of Medicine, Chulalongkorn University, Bangkok 10330, Thailand

**Keywords:** COVID-19, SARS-CoV-2, vaccines, anti-SARS-CoV-2 spike total antibodies, surrogate viral neutralizing antibody, T-cell immune response, CoronaVac, ChAdOx1, BNT162b2, booster

## Abstract

Inactivated SARS-CoV-2 vaccine (CoronaVac) is commonly used in national immunization programs. However, the immune response significantly declines within a few months. Our study assessed the immune response against SARS-CoV-2 after receiving booster shots of BNT162b2 or ChAdOx1 among health care workers who previously received CoronaVac as their primary immunization. Fifty-six participants who received ChAdOx1 and forty-two participants who received BNT162b2 were enrolled into this study, which evaluated immune responses, including anti-SARS-CoV-2 spike total antibodies (Elecsys^®^), surrogated viral neutralization test (sVNT) to ancestral strain (cPass™; GenScript), five variants of concern (Alpha, Beta, Gamma, Delta, and Omicron) (Luminex; multiplex sVNT) and the ELISpot with spike (S1 and S2) peptide pool against the ancestral SARS-CoV-2 strain. The samples were analyzed at baseline, 4, and 12 weeks after primary immunization, as well as 4 and 12 weeks after receiving the booster. This study showed a significant increase in anti-SARS-CoV-2 spike total antibodies, sVNT, and T-cell immune response after the booster, including against the Omicron variant. Immune responses rapidly decreased in the booster group at 12 weeks after booster but were still higher than post-primary vaccination. A fourth dose or a second booster should be recommended, particularly in health care workers.

## 1. Introduction

In 2020, there are many coronavirus diseases 2019 (COVID-19) vaccines that have been used to prevent severe acute respiratory syndrome coronavirus 2 (SARS-CoV-2) infection and decrease the severity of the diseases. CoronaVac (an inactivated SARS-CoV-2 vaccine, Sinovac Life Science) is widely used as primary vaccination. A waning immunity of the primary vaccination (two doses of CoronaVac) over time is observed after 12 weeks, which indicates that a booster dose is needed [1]. Moreover, the emergence of variants of concern (VOCs) raises concerns about the vaccine’s effectiveness, which depending on the VOC. The five major VOCs recognized by the World Health Organization (WHO) are Alpha, Beta, Delta, Gamma, and Omicron [2]. In several studies, vaccine efficacy was found to be lower against Delta and Omicron variants in particular [3,4,5].

Homologous or heterologous COVID-19 prime-boost vaccinations are introduced and reported to improve the humoral and cellular immune responses [6,7]. A homologous third dose of CoronaVac demonstrated increased immune responses against SARS-CoV-2; however, immunogenicity was lower when compared to a heterologous prime-boost with BNT162b2 [8,9,10,11,12]. Heterologous prime-boost vaccination with an mRNA vaccine or a viral vector vaccine is widely used as a booster dose because it can induce a high immune response and higher antibody levels, which are likely to provide greater protection against infection, as well as to overcome the new VOCs when there is a lack of specific VOC COVID-19 vaccines [11,13]. However, data on the dynamic immune response to each VOC after the ChAdOx1 booster or BNT162b2 booster in a CoronaVac-based regimen are limited.

The neutralizing antibody (Nab) titer against SARS-CoV-2 is a highly predictive indicator of protective immunity after vaccination or infection, and several of surrogate virus neutralization tests (sVNTs) are widely used. The neutralizing antibody levels from sVNTs are well correlated to the conventional live virus-neutralizing test or pseudovirus-mediated viral neutralization test [14]. However, sVNT needs to have a specific target to the viral spike protein receptor-binding domain (RBD) of each VOC; therefore, a multiplex sVNT platform was developed that was reported to be highly correlated with the live virus-neutralizing test. Additionally, the binding antibody against the SARS-CoV-2 can be used to monitor immune response.

In this study, we describe the dynamics of the immune responses (both antibody and T-cell) against SARS-CoV-2 ancestral type and VOCs in healthy adults who had received a primary immunization with inactivated SARS-CoV-2 (CoronaVac) and a booster with either ChAdOx1 or BNT162b2 at 4 and 12 weeks after receiving the booster.

## 2. Materials and Methods

### 2.1. Study Design and Participants

A prospective cohort study was conducted among health care workers (HCWs) aged 18 years or older who received CoronaVac (3–4 weeks apart) as their primary vaccination (2 doses of CoronaVac) at a tertiary care center in Bangkok, Thailand, during March-April 2021 and have results of their immune responses at 4 and 12 weeks after primary vaccination [1]. At 3 months after primary vaccination, the participants were non-randomized to voluntarily receive their booster dose of either ChAdOx1 or BNT162b2.

The exclusion criteria at enrollment were ongoing immunosuppressive medication, any vaccination within 1 month, and having received any blood components or intravenous immunoglobulin within 3 months. The termination criteria were either SARS-CoV-2 infection or having received prophylaxis or investigational treatment against COVID-19.

### 2.2. Study Procedures

#### 2.2.1. Demographics and Clinical Data

Baseline demographics and clinical data, such as medical history, current medications used, history of exposure to COVID-19 patients, and history of SARS-CoV-2 infection were collected. All HCWs were highly aware of their infection risk. Moreover, they had to follow the surveillance protocol, which included completing a daily questionnaire to screen for COVID-19 symptoms and exposure risk. Any HCWs who had any symptoms or risk factors were tested for SARS-CoV-2 infection using the RT-PCR technique.

#### 2.2.2. Sample Collection and Study Protocol

For antibody response assessment, 4 mL of clotted blood was collected at 4 (+/−1 week) and 12 weeks (+/−2 weeks) after primary vaccination, as well as 4 (+/−1 week) and 12 weeks (+/−2 weeks) after the booster dose with either ChAdOx1 or BNT162b2. For T-cell response assessment, peripheral blood mononuclear cells (PBMCs) from whole blood were isolated by density gradient centrifugation using Lymphoprep™ (STEMCELL Technologies, Vancouver, Canada) at 1500 rpm for 30 min at room temperature. Isolated PBMCs were washed with RPMI 1640 medium (Gibco, Life Technologies Corporation, NY, USA), which was supplemented with 10% fetal bovine serum (FBS). The cells were then cryopreserved in 10% dimethyl sulfoxide (DMSO) in FBS for further experiments.

#### 2.2.3. Assessment of Antibody Responses against SARS-CoV-2 

The antibody titers against SARS-CoV-2 were determined by anti-SARS-CoV-2 spike total antibodies using the ELISA technique (Elecsys^®^), surrogate neutralizing antibody to ancestral type (cPass^TM^, GenScript), and Multiplex sVNT (Luminex) against five VOCs (Alpha, Delta, Beta, Gamma, and Omicron).
Anti-SARS-CoV-2 spike total antibodies

Anti-SARS-CoV-2 spike total antibodies were detected by Elecsys^®^ Anti-SARS-CoV-2 S using Cobas e411 immunoassay analyzers (Roche Diagnostics, Rotkreuz, Switzerland), which is also used by the US Food and Drug Administration (FDA) under the Emergency Use Authorization (EUA) Act. Elecsys^®^ (Roche Diagnostics, Rotkreuz, Switzerland) is the immunoassay used to detect total SARS-CoV-2 antibodies against the RBD of the S antigen, and the antibody level is reported as U/mL. This assay has a measuring range of 0.40–250 U/mL. A volume of 200 mL of 100-fold diluted sample was used following the manufacturer’s protocol (measuring range, 40–25,000 U/mL). For the participants who had anti-SARS-CoV-2 spike total antibodies over the maximum measuring range (>25,000 U/mL), the samples were further diluted to 1000-fold and re-evaluated using an Elecsys^®^ Diluent universal kit [15]. The assigned U/mL is equivalent to binding antibody units (BAU/mL), as defined by the WHO International Standard for anti-SARS-CoV-2 immunoglobulin (NIBSC code 20/136). Results reported in U/mL do not need to be converted to another unit and can be directly compared to the results of other studies using BAU/mL [16].
2.SARS-CoV-2 neutralizing antibodies to ancestral type

The cPass™ SARS-CoV-2 neutralization antibody detection kit (GenScript, Piscataway, NJ, USA) measures SARS-CoV-2-neutralizing antibodies that block recombinant SARS-CoV-2 RBD conjugated with horseradish peroxidase (HRP-RBD) from binding to the human ACE2 receptor protein (hACE2). The neutralizing antibody level was detected according to the manufacturer’s instructions. Serum samples and controls were diluted to 1:10 and pre-incubated with HRP-RBD for 30 min at 37 °C, allowing SARS-CoV-2-neutralizing antibodies to bind to HRP-RBD. The mixture was then added to a microplate pre-coated with hACE2, where the unbound HRP-RBD and HRP-RBD-bound non-neutralizing antibodies were captured. HRP-RBD complexed with neutralizing antibodies was subsequently removed during the washing steps. After washing, 3,3′,5,5′-tetramethylbenzidine (TMB) substrate was added, followed by the stop solution to terminate the reaction, and the optical density (OD) of the reaction was measured at 450 nm. The antibody level was reported as the percentage inhibition by using the equation from the manufacturer with a cut-off level for SARS-CoV-2-neutralizing antibody detection of ≥30% inhibition. The seroconversion rate was defined as sVNT ≥ 68%, which was adopted from the US FDA’s guidelines for a high titer of COVID-19 convalescent plasma [17].
3.Multiplex sVNT to ancestral type SARS-CoV-2 and VOCs

The sVNT was adapted using the Luminex platform to perform multiplex sVNT as previously described [18]. His-Avi-Tag, biotin-labeled receptor-binding domain (RBD) from SARS-CoV-2 ancestral strain and 5 VOCs (Alpha, Delta, Beta, Gamma, and Omicron) were coupled to paramagnetic MagPlex-Avidin microspheres from Luminex. Diluted RBD-coated microsphere mixture containing approximately 600 beads per RBD protein was incubated with 1:10 diluted sera in a 1:1 ratio in a 96-well plate for 1 h at 37 °C, under shaking at 800 rpm. Human ACE2 receptor protein phycoerythrin (PE)-conjugated (1 ug/mL, GenScript) was then added to the plate and incubated for 30 min at 37 °C with shaking at 800 rpm. After incubation, the mixture was washed twice with assay buffer (PBS + 1% bovine serum albumin), followed by resuspension in assay buffer. The plate was then read using a Bioplex MAGPIX system (Luminex, Austin, TX, USA) to acquire the data.

#### 2.2.4. Assessment of T-Cell Response against SARS-CoV-2 

SARS-CoV-2-specific T-cell responses were investigated by enzyme-linked immunospot assay (ELISpot assay). ELISpot plates (Millipore) were coated with human IFNγ antibody (1-D1K, Mabtech; 5 μg/mL) overnight at 4 °C. Then, 2.5 × 10^5^ PBMCs were stimulated with overlapping spike (S1 and S2) peptide pool (Genscript, Piscataway NJ, USA) of SARS-CoV-2 in AIM-V medium (Gibco) at a final concentration of 2 ug/mL for 16 h. Negative control and positive control, CMV lysates (Meridian Bioscience, Cincinnati, OH, USA), and phytohemagglutinin (Sigma) were also included. The spots were quantified using an ELISpot analyzer (ImmunoSpot, Cleveland, OH, USA). Spot counts of negative control wells were subtracted from the peptide-stimulated wells to generate normalized results. These are reported as spot forming units (SFU) per million PBMCs. The responses higher than mean SFU counts plus 3 standard deviations (SD) of unstimulated wells, which is higher than 19 SFU per million PBMCs, were considered to be positive [19].

### 2.3. Statistical Analysis

Descriptive statistics were used to present demographic data, as well as clinical and laboratory parameters. Continuous variables were presented as the median and interquartile range (IQR). The Wilcoxon rank sum test was used to compare the continuous variables between the two groups. Chi-square test or Fisher exact test was used to compare the proportion between groups. Paired t test was used to compare the change in immunogenicity at week 12 and week 4 (reference) after the vaccine booster dose within group and was reported as geometric mean (GM) and geometric mean ratio (GMR) with 95% confidence interval (CI). Factors associated with anti-SARS-CoV-2 spike total antibody levels after the booster dose were analyzed by random-effects generalized least square regression (log transform). The correlation between anti-SARS-CoV-2 spike total antibodies and other immune responses was determined by the Spearman rank test. Statistical significance was defined as *p*-value < 0.05. STATA version 15.1 (Stata Corp., College Station, TX, USA) was used for statistical analysis.

## 3. Results

### 3.1. Participant Characteristics 

From July to August 2021, 56 participants received the ChAdOx1 booster and 42 participants received the BNT162b2 booster. Blood was collected after 4 and 12 weeks post booster. The median age of participants who had CoronaVac was 41 (31–52) years, and most of them were female (Table 1). The median (IQR) interval after the second CoronaVac and ChAdOx1 booster was 88 (74–92) days and 113 (112–115) days for the BNT162b2 booster. Fourteen participants had a history of contact with COVID-19 patients without appropriate personal protective equipment and/or had symptoms suspected to be of COVID-19, and one participant was diagnosed with COVID-19 3 days after receiving the booster; the latter participant was excluded from the analysis. This infected participant had mild severity of the disease, with upper respiratory tract symptoms and no pneumonia.

### 3.2. Immune Response

#### 3.2.1. Anti-SARS-CoV-2 Spike Total Antibodies

Anti-SARS-CoV-2 spike total antibodies after primary immunization and booster dose are shown in Table 2. Anti-SARS-CoV-2 spike total antibodies at 12 weeks after primary vaccination (2 doses of CoronaVac) were lower than after 4 weeks in both groups. After the participants received the ChAdOx1 booster, the anti-SARS-CoV-2 spike total antibodies at 4 and 12 weeks post booster were significantly lower than in the BNT162b2 booster group, *p* < 0.001. Anti-SARS-CoV-2 spike total antibodies rapidly decreased in the BNT162b2 booster group 12 weeks after booster (79% in the BNT162b2 group vs. 59% in the ChAdOx1 group) but were still higher than post-primary vaccination.

#### 3.2.2. Surrogate Viral Neutralizing Antibody to Ancestral Type by GenScript and VOCs by Luminex

The sVNT to ancestral type at 12 weeks after primary vaccination (2 doses of CoronaVac) was lower than at 4 weeks in both groups. The median (IQR) of sVNT to ancestral type after booster with ChAdOx1 was significantly lower than the BNT162b2 booster group at 4 weeks after the booster (Table 2 and Figure 1). sVNT to each VOC was different. Overall, the median sVNT to each VOC in the BNT162b2 booster group was higher than the in ChAdOx1 booster group. In addition, the median of sVNT to wild type, Alpha, and Delta had similar patterns, whereas the Beta and Gamma variants had similar patterns. However, sVNT to Omicron was significantly lower than that of the other VOCs (Figure 2 and Appendix A). The median (IQR) of sVNT to Omicron after booster with ChAdOx1 at 12 weeks was 26.6% (12.4–37.7%), which is significantly lower than booster with BNT162b2 (median 50.7 (IQR 34.6–71.9%), *p* < 0.001. The decline rate of sVNT to Omicron from week 4 to week 12 was 45% among BNT162b2 and 62% among ChAdOx1 participants; however, the decline rate of sVNT to other VOCs was only 6–20%. (Table 2 and Table 3).

According to the US FDA guidelines for a high titer of COVID-19 convalescent plasma, the % inhibition using the cPass^TM^ SARS-CoV-2 Neutralization Antibody Detection Kit should be ≥68% [17]. None of the participants had sVNT > 68% to Omicron after primary CoronaVac immunization; however, the seroconversion rate increased to 83% and 35% after 4 weeks of BNT 162b2 and ChAdOx1 boosters, respectively. The seroconversion rate of other VOCs, except Omicron, was 94–100% in the BNT162b2 group and 80–100% in the ChAdOx1 group.

#### 3.2.3. Assessment of T-Cell Response by ELISpot

The results of IFN-γ secreting T cell by Pool S1 and Pool S2 are shown in Figure 3 and Appendix A. The responses had a similar pattern, which showed that there was a reduction in the immune response 12 weeks after primary immunization, increased immune response 4 weeks after the booster dose, and a reduction in the immune response 12 weeks after the booster. Percent positive responses after the booster at week 4 and week 12 were 100% and 90% with BNT162b2 and 80% and 80% with ChAdOx1, respectively. Although there was no significant difference between the BNT162b2 and ChAdOx1 booster groups, the median SFU after BNT162b2 was higher than after ChAdOx1 4 and 12 weeks post booster.

## 4. Discussion

This prospective observational cohort study reported the dynamics of antibodies against ancestral type and five VOCs, as well as T-cell responses in participants who received a primary immunization with CoronaVac and a booster with BNT162b2 or ChAdOx1. The results showed that there was a decrease in the immune response 12 weeks after primary immunization. Since November 2021, the variant has changed to Omicron, and a booster dose is recommended. At 4 weeks post booster, both BNT162b2 and ChAdOx1 resulted in significant boost of anti-SARS-CoV-2 spike total antibodies. The boost achieved by BNT162b is approximately 4 times higher than achieved by ChAdOx1, which correlates with higher sVNT against the Omicron strain (median 80.9% versus 54.5%). At 12 weeks post booster, anti-RBD rapidly declined which may reduce the effectiveness of the vaccines in preventing infection. Cell-mediated immune response measured by ELISpot assay showed that there was a similar T-cell immune response after booster in the ChAdOx1 group and BNT162b2 booster group. Cell-mediated immunity has been reported as an immune response that reduces the severity of disease [20].

Among the ChAdOx1 booster group, the anti-SARS-CoV-2 spike total antibodies to ancestral type SARS-CoV-2 and sVNT to ancestral type, Alpha, Beta, and Delta at 4 weeks post booster in this study were similar to the results of other studies [13,21]. However, the median of sVNT to omicron was 61.2% at 4 weeks post booster but declined to 26.6% at 12 weeks post booster, which indicates that one booster may not be able to prevent infection and, hence, an additional booster is needed. Among the BNT162b2 booster group, the immune response by neutralizing antibody to ancestral-type SARS-CoV-2, Delta, and Omicron at 4 weeks post booster was similar to that reported in other studies [10,22,23]. However, there was a significant decline in sVNT to omicron at 12 weeks after booster (median (IQR) sVNT to Omicron, 50.7% (IQR 34.6–71.9%)). The immune response in the BNT162b2 booster group was higher than that in the ChAdOx1 booster group 4 weeks post booster, which is in line with data from a previous study conducted in Brazil [11]. The mRNA vaccines have demonstrated a high neutralizing antibody response when compared to other vaccine platforms, including viral vector vaccines or inactivated virus vaccines used in primary immunization [24].

Interestingly, at weeks 12 after booster, the level of sVNT to ancestral type (cPass^TM^) was not as drastically decreased as the level of anti-SARS-CoV-2 spike total antibodies. This might be because the anti-SARS-CoV-2 spike total antibodies include non-neutralizing antibodies, which might have a different rate of decay. In this study, we used a commercial sVNT kit, which included test samples and control sera tested using a fixed dilution of 1:10. It is clear that all sera after booster have a high level of NAbs, reaching the saturation level (>95%) of NAbs dateable by this kit, as shown in our previous studies [25]. However, the sVNT of VOCs by multiplex sVNT (Luminex) detected a decrease in sVNT from week 4 to week 12.

T-cell responses to the spike protein increased substantially after a booster dose with either ChAdOx1 or BNT162b2. The response rate and the magnitude of T-cell response seem to be higher in the BNT162b2 booster group compared to the ChAdOx1 group, which supports the findings of a previous study [26]. It should be noted that there were some positive T-cell responses at baseline similar to a previous report [27]. However, after the primary vaccination (two doses of CoronaVac), the magnitude of T-cell responses was higher in every case, supporting the need for immunization. Although we did not measure T-cell responses to pool peptides derived from variant strains, the T-cell responses to Omicron should be reserved, as previously shown. [28,29] There was a correlation between specific T-cells and antibody responses to the spike protein (Supplement correlation graph), which is similar to the immune response seen in patients with primary infection [30].

The difference in the humoral immune response to each VOC occurred because the vaccines were designed to recognize the ancestral-type spike protein of SARS-CoV-2. Delta has 9 mutations, and Omicron has at least 46 highly prevalent mutations, which is more than any previous variant belonging to the S protein; therefore, it is not surprising to see a decrease in the humoral immune response toward other variants [4,31]. Whereas neutralizing antibodies target the spike protein and block cellular entry, the cytotoxic T cell plays an important role in eliminating virus-infected cells. The heterologous prime-boost strategy with different platforms has been shown to effectively improve immunogenicity [32]. The mRNA vaccine seems to induce a stronger immune response than the viral vector vaccine; however, the immune response to Omicron rapidly declines, which indicates that an additional booster dose is needed, first for health care workers, who are on the frontline in this pandemic, and then for all populations. A second booster should be an mRNA vaccine or a protein-based vaccine with adjuvant. For example, a protein-based vaccine with an adjuvant from Novavax shows promising results; it has high anti-RBD at the primary vaccination or after a booster [33,34].

This study has several strengths. First, it is a longitudinal cohort study. Second, this study includes antibody and T-cell response data from baseline to primary immunization, as well as post booster. Moreover, this study includes immune response data to all VOCs, especially the Delta and Omicron variants, which were predominantly circulating around the world at the time when the study was conducted. There are some limitations in this study. First, the study was non-randomized; therefore, a difference in age was reported between groups. However, we analyzed factors associated with anti-SARS-CoV-2 spike total antibodies (Appendix A) and reported that the type of booster vaccine is significantly associated with this level (adjusted ratio 3.61 (95%CI 1.11–11.76, *p* 0.03)), whereas age and anti-SARS-CoV-2 spike antibodies prior to booster were not associated with this level. (Adjusted ratio 0.99 (95%CI 0.98–1.01, *p* 0.364, adjusted ratio 1.01 (95%CI 0.98–1.04, *p* 0.487)). Second, the sample size was small, so there was not enough power to study the clinical efficacy of the vaccine regimens. Lastly, the T-cell responses were only performed in a subset of the sample, and in this study, we did not test the T-cell responses to the spikes from different variants. However, a recent study showed that T-cell responses to the spikes of various variants were still detectable among vaccinated individuals [28,29,35].

## 5. Conclusions

The heterologous prime-boost regimen with CoronaVac followed by a booster with BNT162b2 or ChAdOx1 induced a significant increase in anti-SARS-CoV-2 spike total antibodies, neutralizing antibodies, and T cell responses. The BNT162b2 group had higher neutralizing antibodies against the Omicron strain, which is the predominant strain in the year 2022. The humoral immune response tends to decrease rapidly after 12 weeks post booster; therefore, health care workers, who are at high risk of virus acquisition, may need an additional booster dose (i.e., second booster) 3 months after the first booster dose. Future studies should assess the long-term immune response to determine the appropriate time to administer the second booster.

## Figures and Tables

**Figure 1 vaccines-10-00639-f001:**
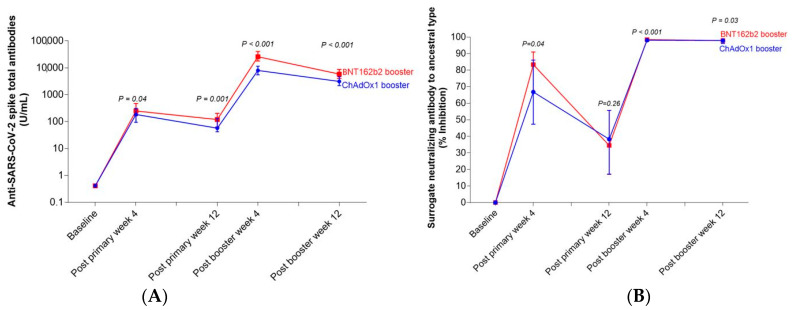
Dynamic of immune response of participants who received primary CoronaVac vaccination and were boosted with BNT162b2 or ChAdOx1 at week 12 for (**A**) anti-SARS-CoV-2 spike total antibodies (U/mL) by Elecsys and (**B**) surrogate neutralizing antibody to ancestral type (% inhibition) by GenScript at 4 and 12 weeks post booster.

**Figure 2 vaccines-10-00639-f002:**
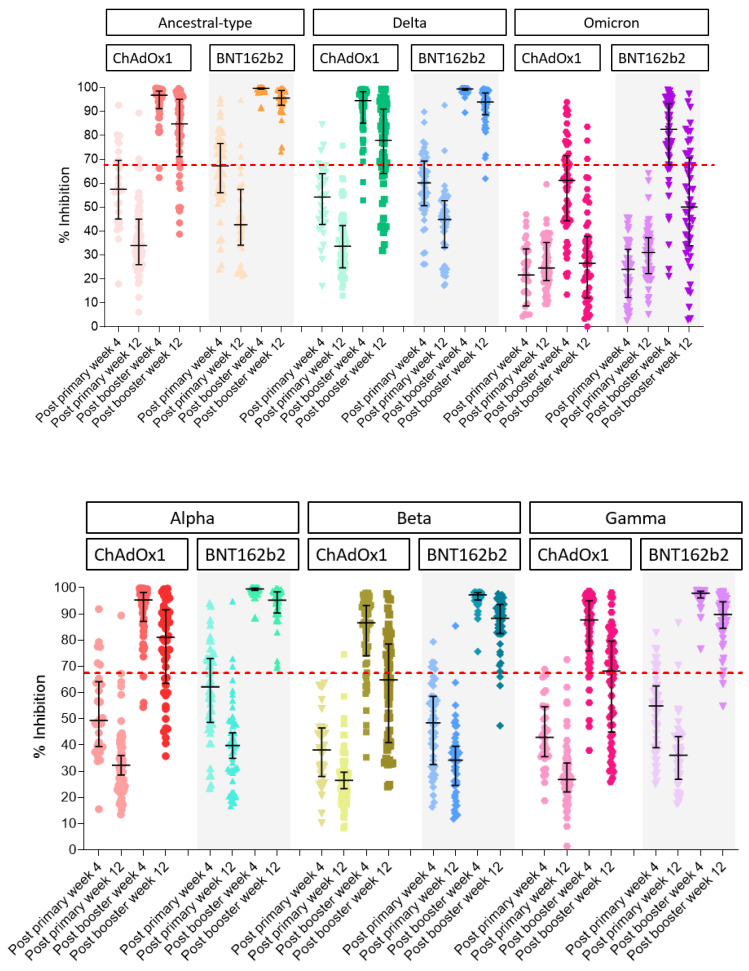
Dynamics of immune response surrogate neutralizing antibody (sVNT) to ancestral type, Delta, Omicron, Alpha, Beta, and Gamma in participants who received primary CoronaVac vaccination and were boosted with BNT162b2 or ChAdOx1 by multiplex sVNT.

**Figure 3 vaccines-10-00639-f003:**
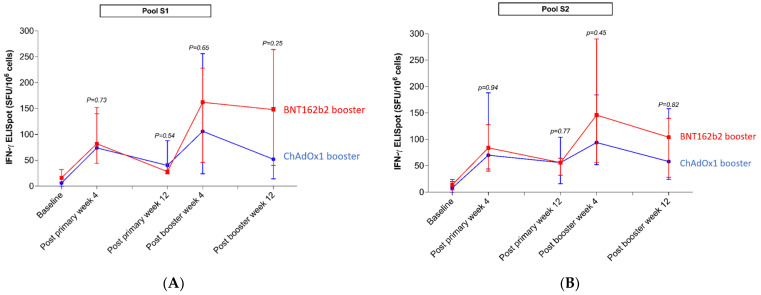
Dynamics of T-cell response by ELISpot (**A**) Pool S1, and (**B**) Pool S2 in participants who received primary CoronaVac vaccination and were boosted with BNT162b2 or ChAdOx1.

**Table 1 vaccines-10-00639-t001:** Baseline characteristics of the participants who received primary immunization with CoronaVac and booster with ChAdOx1 or BNT162b2.

Characteristics	ChAdOx1 Booster	BNT 162b2 Booster	*p*-Value
	N = 56	N = 42	
**Age, years, median (IQR)**	47 (34–53)	32 (30–45)	0.001
**Age group, n****(**%**)**			0.01
**20–30 years**	7 (12.5)	15 (35.7)	
**31–40 years**	12 (21.4)	14 (33.3)	
**41–50 years**	13 (23.2)	5 (11.9)	
**51–60 years**	24 (42.9)	8 (19.1)	
**Female, n****(**%**)**	44 (78.6)	34 (81)	0.005
**BMI (kg/m^2^)**, **median (IQR)**	22.6 (20.3–25.9)	21.5 (20–25.8)	0.45

IQR, interquartile range; BMI, body mass index.

**Table 2 vaccines-10-00639-t002:** Comparison of anti-SARS-CoV-2 spike total antibodies (U/mL) by Elecsys and surrogate viral neutralizing antibody to ancestral strain by GenScript in participants who received primary immunization with CoronaVac and booster with ChAdOx1 or BNT 162b2.

Median (IQR)	Anti-SARS-CoV-2 Spike Total Antibodies (U/mL)			Surrogate Viral Neutralizing Antibody to Ancestral Type (%Inhibition) by GenScript		
**Primary immunization**	**(N = 56)**	**(N = 42)**	** *p* ** **-value**	**(N = 56)**	**(N = 42)**	***p*-value**
Week 4	179 (92–301)	242 (163–462)	0.04	66.8 (47.4–86.1)	83.4 (66.8–91.0)	0.04
Week 12	56 (41–126)	117 (61–198)	0.001	39.4 (18.6–57.8)	41.3 (31.7–59.2)	0.26
**Booster**	**ChAdOx1**	**BNT162b2**	** *p* ** **-value**	**ChAdOx1**	**BNT162b2**	***p*-value**
Week 4	7768 (5349–11,142)	25,129 (17,531–39,434)	<0.001	98.1 (97.9–98.2)	98.5 (98.5–98.6)	<0.001
Week 12	3139 (2185–4660)	6558 (3836–8885)	<0.001	97.9 (95.8–98.1)	97.9 (97.8–98.2)	0.03

**Table 3 vaccines-10-00639-t003:** Comparison of the changes in anti-SARS-CoV-2 spike total antibodies and surrogate viral neutralizing antibody (sVNT) by multiplex sVNT between week 4 and week 12 after the booster dose.

Post Booster	Primary Immunization with CoronaVac and Booster with ChAdOx1			Primary Immunization with CoronaVac and Booster with BNT162b2		
	GM	GMR	% Decrease	GM	GMR	% Decrease
**Anti-SARS-CoV-2 spike total antibodies**						
week 4	7292.9 (6049.7–8791.6)	ref		29,268.9 (23,921.4–35,812.1)	ref	
week 12	2960.5 (2421.0–3620.2)	0.41 (0.37–0.45)	59%	6200.0 (5043.5–7621.8)	0.21 (0.18–0.25)	79%
**Surrogate viral neutralizing antibody** **(sVNT)**						
**Ancestral type**						
week 4	92.8 (90.3–95.3)	Ref		99.5 (99.4–99.7)	ref	
week 12	79.2 (74.1–84.6)	0.85 (0.81–0.89)	15%	93.8 (91.7–96.0)	0.94 (0.92–0.96)	6%
**Alpha**						
week 4	90.3 (87.0–93.8)	ref		90.3 (87.0–93.8)	ref	
week 12	74.8 (69.4–80.6)	0.83 (0.79–0.87)	17%	74.8 (69.4–80.6)	0.83 (0.79–0.87)	17%
**Beta**						
week 4	79.7 (75.0–84.8)	ref		96.4 (95.7–97.2)	ref	
week 12	58.1 (52.3–64.4)	0.73 (0.68–0.78)	27%	85.2 (81.5–89.0)	0.88 (0.85–0.92)	12%
**Gamma**						
week 4	81.9 (77.4–86.7)	ref		97.1 (96.4–97.8)	ref	
week 12	59.8 (53.8–66.4)	0.73 (0.68–0.78)	27%	86.7 (83.3–90.3)	0.89 (0.86–0.93)	11%
**Delta**						
week 4	89.6 (86.3–93.1)	ref		99.1 (98.9–99.3)	ref	
week 12	71.4 (65.6–77.7)	0.8 (0.75–0.84)	20%	91.1 (88.3–94)	0.92 (0.89–0.95)	8%
**Omicron**						
week 4	54.5 (48.6–61.3)	ref		80.9 (75.1–87.1)	ref	
week 12	20.8 (15.3–28.2)	0.38 (0.29–0.49)	62%	44.4 (35.3–55.7)	0.55 (0.43–0.69)	45%

GM, geometric mean; GMR, geometric mean ratio.

## Data Availability

The supporting data for the findings of this study are available from the corresponding author upon reasonable request.

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
