# Peer review of "Dynamics of Neutralizing Antibody and T-Cell Responses to SARS-CoV-2 and Variants of Concern after Primary Immunization with CoronaVac and Booster with BNT162b2 or ChAdOx1 in Health Care Workers"

_vaccines, 2022, doi:10.3390/vaccines10050639_

Round 1
Reviewer 1 Report
Reviewer’s comments for vaccines-1676922, “Dynamics of neutralizing antibody and T-cell responses to SARS-CoV-2 and variants of concern after primary immunization with CoronaVac and booster with BNT162b2 or ChAdOx1 in health care workers” by Jantarabenjakul et al.
The authors evaluated the T cell and antibody responses following ChAdOx1 or BNT162b2 booster in a cohort who received CoronaVac as primary vaccination. This study highlights the advantage of a heterologous vaccination strategy, with significantly increased anti-RBD titers and neutralizing capacity of antibodies against ancestral type and VOCs. T cell responses were also increased 4 weeks after booster, although no significant difference was observed between ChAdOx1 and BNT162b2 by ELISpot. This study is important as it guides the formulation of vaccination strategies.
Major comments:
- Some comments should be included on how the age difference between the 2 groups (Table 1) can affect the T cell and antibody responses to booster, especially when mounting an adaptive immune response can be a problem due to immune-senescence.
- It will be nice to have age-matched the 2 groups but I understand that can be a problem given the nature of the study.
- Ideally, controls like BNT162b2 or ChAdOx1 primary vaccination can be used as controls, but it can be limiting given the circumstances.
- From the primary CoronaVac immunization, the imminent ChAdOx1 booster group are already developing lower antibody levels compared to the BNT162b2 group, could there be other factors that are reducing the development of antibody responses?
- Can the authors comment on the drop in antibody titers vs neutralizing capacity (sVNT)? From table 2, although the antibody titer is reduced 2-4 fold from 4 weeks to 12 weeks post booster, the sVNT data is not reduced drastically.
Minor comments:
- It will be more appropriate to term the study of antibody responses then than B cell responses (Line 100).
Reviewer 2 Report
To fight against pandemic SARS-CoV-2 infection, many vaccines have been developed. These vaccines are remarkably effective to prevent sever COVID-19. However, some problems are arising. Titers of neutralizing antibodies are decreasing, and several months after vaccination, infection would be poorly inhibited. Another serious problem is the appearance of vaccine-resistant variant strains. The booster vaccination is beneficial because it can increase not only antibody titers but also reactivity to variant viruses. However, dynamics of immune responses including duration, antigen titer, reactivity to ancient and variant viruses, and T cell responses are unclear.
In this manuscript, the authors analyzed immune responses of the HCWs who had received primary vaccination by two doses of CoronaVac followed by booster of either BNT162b2 or ChAdOx1. Data presented were very informative because (1) samples were collected in different time points of vaccination (baseline, 4 and 12 weeks after primary vaccination, and 4 and 12 weeks after booster), (2) two different vaccines (BNT162b2 and ChAdOx1) were compared, (3) neutralization activity not only for ancestral strain but also for five VOCs was analyzed, and finally, (4) in addition to antibody titer (B cell responses), T cell responses were compared.
They concluded that (1) after B cell responses of primary CoronaVac vaccination waned, booster immunization reactivated production of antibody with neutralizing activity against SARS-CoV-2 including VOCs. (2) BNT162b2 was more effective as booster than ChAdOx1. (3) Ab production decreased in 12 weeks to the level which was not enough for protection so that further booster was necessary.
These data are valuable for many people so that it should be published. However, as described below, the conclusion (2) is not fully supported by their data. Therefore, the manuscript cannot be accepted for publication unless the authors have to rewrite the paper.
Major problem:
There was significant difference in primary response between ChAdOx1 and BNT162b2 booster groups. Antibody titers at post primary week 12 was higher in BNT162b2 booster group than ChAdOx1 booster group. It suggested higher primary response in BNT162b2 booster group which might be reflected on higher response after boosting. Additionally, antibody titers at week 12 may not indicate the immune status at the timing of boosting because BNT162b2 group was boosted at day 113 (week 16). These facts may be confounding factors, and should be carefully considered for comparison between ChAdOx1 and BNT162b2 booster. However, it was not discussed in the manuscript.
Relatively minor points:
(1) It seems that information about sample collections and study protocol is insufficient. Were the participants same as those of their previous study Jantarabenjakul et al. (DOI 10.12932/AP-250721-1197)? In that paper, 94 participants were received CoronaVac, however, N=98 in this manuscript (Table 1). Or the participants of these two works were different?
Was the study protocol including booster designed at the beginning, i.e. before primary vaccination? Or the protocol of boosting was designed after the analysis of primary responses?
And when the baseline samples were collected?
(2) In line 314, I could not understand why protein-based vaccine with adjuvant was recommended for second booster. References 30 and 31 do not mention Omicron.
(3) In line 327, about T cell response to VOC, the author can discuss by citing recent publications such as Keeton et al., Nature 603: 488-492 (doi.org/10.1038/s41586-022-04460-3) and Liu et al., Nature 603: 493-496 (doi.org/10.1038/s41586-022-04465-y)
(4) Line 208, I cannot find 79% or 59% in Tables. Line 270, what does 4 times mean?
(5) Line 167, which is the supplier of ImmunoSpot?
Round 2
Reviewer 2 Report
The authors carefully responded to my comments and revised the manuscript. I do not have other questions.
I found two minor errors to be corrected.
Table 1, Total number of the participants in age groups was 57, not 56.
Lines 297 and 298, Nab should be NAb.